# A Two-Dimensional Precision Level for Real-Time Measurement Based on Zoom Fast Fourier Transform

**DOI:** 10.3390/mi14112028

**Published:** 2023-10-30

**Authors:** Haijin Fu, Zheng Wang, Xionglei Lin, Xu Xing, Ruitao Yang, Hongxing Yang, Pengcheng Hu, Xuemei Ding, Liang Yu

**Affiliations:** 1Ultra-Precision Optoelectronic Instrument Engineering Center, School of Instrument Science and Engineering, Harbin Institute of Technology, Harbin 150080, China; haijinfu@hit.edu.cn (H.F.); 18b901009@stu.hit.edu.cn (Z.W.); linxl@stu.hit.edu.cn (X.L.); xingxu@hit.edu.cn (X.X.); ruitao.yang@hit.edu.cn (R.Y.); takeon@126.com (H.Y.); hupc@hit.edu.cn (P.H.); xmding@hit.edu.cn (X.D.); 2Key Laboratory of Ultra-Precision Intelligent Instrumentation, Harbin Institute of Technology, Ministry of Industry and Information Technology, Harbin 150080, China

**Keywords:** precision level, homodyne interference, zoom FFT, real-time angle measurement

## Abstract

This paper proposes a two-dimensional precision level for real-time measurement using a zoom fast Fourier transform (zoom FFT)-based decoupling algorithm that was developed and integrated in an FPGA. This algorithm solves the contradiction between obtaining high resolution and obtaining high measurement speed, and achieves both high angle-resolution measurement and real-time measurement. The proposed level adopts a silicone-oil surface as the angle-sensitive interface and combines the principle of homodyne interference. By analyzing the frequency of the interference fringes, the angle variation can be determined. The zoom-FFT-based decoupling algorithm improves the system’s frequency resolution of the interference fringes, thereby significantly enhancing the angle resolution. Furthermore, this algorithm improves the efficiency of angle decoupling, while the angle decoupling process can also be transplanted to the board to realize real-time measurement of the level. Finally, a prototype based on the level principle was tested to validate the effectiveness of the proposed method. The principle analysis and test results showed that the angle resolution of the prototype improved from 9 arcsec to about 0.1 arcsec using this angle-solution method. At the same time, the measurement repeatability of the prototype was approximately ±0.2 arcsec. In comparison with a commercial autocollimator, the angle measurement accuracy reached ±0.6 arcsec.

## 1. Introduction

Precision levels, as an essential instrument for angle error measurement, are widely used in precision-measurement and manufacturing fields, such as in ultra-precision machine tools, nano/micro-coordinate measuring machines (CMMs), and atomic force microscopes [1,2,3]. Moreover, precision levels play a significant role in multi-degree-of-freedom motion error measurement systems [4]. Common types of levels are classified based on different measurement principles, including pendulum-type, bubble-type, fiber-optic-based, and liquid-based levels.

For pendulum-type levels, the structure relies on gravity as a reference and utilizes non-contact sensors to detect the swinging of the pendulum for angle measurement. For instance, researchers at the National Taiwan University proposed a dual-axis inclinometer using a commercial DVD pickup head and a double-layer pendulum, achieving angle measurements within ±30 arcsec with an error better than ±0.7 arcsec [5]. In addition, there are measurement methods using capacitive sensors [6,7]. Although pendulum-type levels can achieve high accuracy, their structural complexity is a limitation.

Bubble-type levels measure angles by observing the position change of bubbles on a liquid surface. Ordinary bubble-type levels rely on visual observation, resulting in low precision, and are only suitable for initial instrument leveling. Advanced bubble-type levels incorporate optoelectronic sensors to detect bubble positions. The multi-axis bubble-level device presented in a U.S. patent [8] and Brooks Automation’s electronic liquid-level wafer [9] are some examples. While these methods have a simple structure, their accuracy remains low.

Fiber-optic levels are typically based on techniques such as fiber Bragg gratings [10], phase-shifted fiber Bragg gratings [11], and Fabry–Perot interferometers [12,13]. These methods mainly utilize wavelength-shift detection of Bragg fibers to obtain the measurement angle or employ two Fabry–Perot interferometers to establish the optical Vernier effect for angle measurement. Although these methods generally offer advantages such as simplicity, high sensitivity, and immunity to electromagnetic interference (EMI), further research is still needed to address their nonlinear issues.

Liquid-based levels are usually based on the refraction or reflection of a liquid and combine it with the autocollimation principle to detect the position change of light spots to measure the angle change. Yubin Huang et al. from the Dalian University of Technology proposed a dual-axis level based on transmittance and refraction [14]. This method employs liquid as a reflective medium, achieving errors that are better than ±0.6 arcsec within a measurement range of ±100 arcsec and better than ±5 arcsec within a range of ±800 arcsec. Jingsyan Torng et al. from the Taoyuan Innovation Institute of Technology presented a refractive-based dual-axis level [15], incorporating a reflecting mirror at the bottom of the liquid. After calibration, this method achieves errors better than ±0.4 arcsec within a measurement range of ±30 arcsec and better than ±0.7 arcsec within a range of ±100 arcsec. Other methods include an optoelectronic liquid level based on dual-layer liquid refraction proposed by researchers from Hefei University of Technology [16], an optoelectronic liquid level based on liquid-surface reflection developed by researchers from Tianjin University [4], and a system for measuring laser-beam deviation from the vertical direction, proposed by researchers from Harbin Institute of Technology [17]. These methods all integrate the autocollimation principle, and although measurement accuracy at the level of sub-arcseconds is obtained, issues such as light spot drift still need to be addressed.

Previously, our team proposed a dual-axis optoelectronic level based on wave-front interference fringes [18], which can achieve an angle resolution at the sub-microradian level. This level obtained the measured angle by solving the interference fringes. However, to improve the angle resolution, the results must be obtained offline using a computer, and real-time measurement can hardly be achieved due to the use of zero-padding and curve-fitting algorithms, which occupy a considerable amount of computing resources. In addition, other methods of angle measurement based on interference fringes [19,20,21,22,23] also face such a challenge, that is, the contradiction between high measurement speed and high measurement resolution.

In this paper, we propose a two-dimensional precision level for real-time measurement using a zoom-FFT-based decoupling algorithm developed and integrated in an FPGA, to realize both high angle-resolution measurement and real-time measurement. This decoupling algorithm breaks through the limitation inherent in current hardware and significantly improves the resolution of the level without changing the number and size of CMOS camera pixels. Compared to zero-padding and curve-fitting algorithms, this algorithm does not change the amount of data in the process of solving the contradiction between resolution and measurement speed. Therefore, the algorithm can be integrated into an FPGA, making online measurement possible. In addition, for the level previously proposed by our team, we have simplified its structure and reduced the number of optical components required, making it easier to integrate. Finally, the feasibility of the proposed measurement method was theoretically analyzed, and corresponding tests were conducted to verify its resolution, repeatability, and measurement accuracy.

## 2. Principles and Methods

### 2.1. Structure and Angle Measurement Method of Level

The optical structure of the level is shown in Figure 1. The level utilizes a He–Ne frequency-stabilized laser as the light source, silicone oil as the liquid mirror, a flat mirror as the reference mirror, and a complementary metal-oxide semiconductor (CMOS) industrial camera as the detector for receiving the interference signal. The laser is connected to the level through an optical fiber and emits a beam of linearly polarized light through a collimator in the level. The polarized light is split into two beams by a beam splitter (BS). One beam is the reference light, which is successively reflected by the reference mirror (Mr) and the BS before being received by the camera. The other beam is the measurement light, which is reflected by the liquid mirror and then transmitted through the BS before being received by the camera. In this system, the reference mirror has an initial angle *α*, which introduces an optical path difference and forms the interference fringes on the camera’s receiving surface. This is the basis for angle measurement in the level. Since the reflectivity of the silicone-oil surface is only 3%, the reflectivity of the selected reference mirror is also set to be about 3% to allow the interference fringes to have a high contrast. In addition, the selected CMOS camera has a high sensitivity, so the intensity of the light reflected by the silicone oil and the reference mirror can satisfy the measurement requirements. When the level undergoes a certain angular deviation, the optical path difference between the reference light and the measurement light changes, resulting in variations in the interference fringes.

To accurately establish the relationship between variations in interference fringes and the measured angle, we employ vector-based beam tracing and analyze the system accordingly. Figure 2 depicts the beam-tracing model. To facilitate calculation, the reference path is mirror-symmetric, with the beam-splitter plane as the center, so that it is in the same direction as the measuring path. After being reflected by the beam-splitter plane, the reference light and the measurement light intersect with the reference mirror and the measurement mirror at A (*a*, *b*, *c*) and B (*d*, *e*, *f*), respectively. The two beams eventually intersect at P (*x*, *y*, *L*) on the detector plane. *M_rp_* and *M_mp_* represent the ideal positions of the reference mirror and the measurement mirror, respectively, while *α_r_* and *α_m_* denote the angular deviations of their actual positions. L represents the distance between the detector plane and the ideal position of the reference mirror, and K denotes the distance between the ideal positions of the two mirrors. ${\overset{⃑}{l}}_{r}$ is the normal vector of *M_r_*, and *α_r_* is the initial declination angle of *M_r_*, where ${\overset{⃑}{l}}_{r}=(V_{\mathrm{rx}},V_{\mathrm{ry}},\;1)$. Then, the following relationship is satisfied:(1)$$\tan\alpha_{r}=\sqrt{V_{rx}{}^{2}+V_{ry}{}^{2}}$$

Equation (1) shows that there is a correspondence between the deflection angle *α_r_* and the normal vector of *M_r_*; thus, *V_rx_* and *V_ry_* are used to approximately represent the deflection angles around the Y-axis and the X-axis, respectively. As shown in Figure 2, the point O (*0*, *0*, *0*) is the origin of the coordinate system, and C (*x*, *y*, *u*) represents the projection of P onto the reference mirror. *L_a_* denotes the distance from P to C. According to the geometric relationships, the vector relationship can be obtained as $\overset{⃑}{\mathrm{OC}}⊥{\overset{⃑}{l}}_{r}$. Therefore, the following relationship can be derived:(2)$$xV_{rx}+yV_{ry}+u=0$$
(3)$$L_{a}=L-u=L+xV_{rx}+yV_{ry}$$

Similarly, it can be inferred that the normal vector of the measurement mirror is ${\overset{⃑}{l}}_{m}=(V_{\mathrm{mx}},V_{\mathrm{my}},\;1)$, and the vector relationship can be obtained as $\overset{⃑}{\mathrm{OD}}⊥{\overset{⃑}{l}}_{m}$. *L_b_* represents the distance from the interference point on the detector to the measurement mirror. Given that *O*′ (0, 0, −*K*) and *D* (*x*, *y*, *v*), the following relationship can be obtained:(4)$$xV_{mx}+yV_{my}+v+K=0$$
(5)$$L_{b}=K-v=L+K+xV_{mx}+yV_{my}$$

*M* represents the distance from the light source to the beam-splitter plane, and *N* denotes the distance from the beam-splitter plane to the ideal position of the reference mirror. *L*′ is the distance between *A* and *P*. Therefore, the optical path of the reference light can be expressed as follows (the refractive index of air is n = 1):(6)$$L_{r}=M+N-c+L^{\prime }$$

To obtain the expression for c, ${\overset{⃑}{h}}_{r}=(l,m,i)$ is used to represent the direction vector of the reference reflected light and ${\overset{⃑}{I}}_{r}=(0,0,\;1)$ is used to represent the direction vector of the incident light ray. According to the law of reflection, it can be deduced that $\overset{⃑}{I_{r}}+\overset{⃑}{I_{r}}∥{\overset{⃑}{l}}_{r}$, $\overset{⃑}{h_{r}}-\overset{⃑}{I_{r}}⊥{\overset{⃑}{l}}_{r}$. Consequently, the following relationship can be obtained:(7)$$\frac{l}{V_{rx}}=\frac{m}{V_{ry}}=\frac{i+1}{1}$$
(8)$$lV_{rx}+mV_{ry}+i-1=0$$

According to Equations (7) and (8), the direction vector of the reference reflected light can be expressed as follows:(9)$$\vec{h_{r}}=\left(\frac{2V_{rx}}{1+V_{rx}^{2}+V_{ry}^{2}},\frac{2V_{rx}}{1+V_{rx}^{2}+V_{ry}^{2}},\frac{1-V_{rx}^{2}-V_{ry}^{2}}{1+V_{rx}^{2}+V_{ry}^{2}}\right)$$

Because $\overset{⃑}{h_{r}}∥\overset{⃑}{\mathrm{AP}}$ and $\overset{⃑}{l_{r}}⊥\overset{⃑}{\mathrm{OA}}$, the expression for the vertical coordinate *c* can be deduced as follows:(10)$$c=L+L_{a}-\frac{2L_{a}}{1+V_{rx}^{2}+V_{ry}^{2}}$$

As shown in Figure 2, because $\overset{⃑}{l_{r}}∥\overset{⃑}{\mathrm{PQ}}$ and PQ∥AQ, PQ is the perpendicular bisector of ΔAPC. Consequently, it can be inferred that ΔAPC forms an isosceles triangle, leading to $L^{\prime }=\mathrm{AP}=\mathrm{PC}=L_{a}$. Therefore, the optical path of the reference light is estimated as follows:(11)$$L_{r}=M+N-L+\frac{2L_{a}}{1+V_{rx}^{2}+V_{ry}^{2}}$$

The optical path derivation of the measurement light is similar to that of the reference light. *L*″ is the distance between *B* and *P*, and the optical path of the measurement light can be expressed as follows (the refractive index of air is n = 1):(12)$$L_{m}=M+N-f+L^{\prime \prime }$$

Similarly, the direction vector ${\overset{⃑}{h}}_{m}$ of the measured reflected light can be determined based on Equations (6) to (8), and the vertical coordinate *f* of *B* can be obtained based on the geometric relationships. Given that the point *O*′ (0, 0, −*K*) is known, the following relationship can be derived:(13)$$f=L+L_{b}-\frac{2L_{b}}{1+V_{mx}^{2}+V_{my}^{2}}$$

Since $L^{\prime \prime }=\mathrm{BP}=\mathrm{DP}=L_{b}$, by simultaneously applying Equations (12) and (13), the optical path of the measurement light can be obtained as follows:(14)$$L_{m}=M+N-L+\frac{2L_{b}}{1+V_{mx}^{2}+V_{my}^{2}}$$

In summary, by combining Equations (11) and (12), the following relationship can be obtained:(15)$$\begin{array}{cc} \Delta L & =L_{m}-L_{r}\\ & =\frac{2K}{1+V_{mx}^{2}+V_{my}^{2}}+2L\left(\frac{1}{1+V_{mx}^{2}+V_{my}^{2}}-\frac{1}{1+V_{rx}^{2}+V_{ry}^{2}}\right)\\ & +x\left(\frac{2V_{mx}}{1+V_{mx}^{2}+V_{my}^{2}}-\frac{2V_{rx}}{1+V_{rx}^{2}+V_{ry}^{2}}\right)\\ & +y\left(\frac{2V_{my}}{1+V_{mx}^{2}+V_{my}^{2}}-\frac{2V_{ry}}{1+V_{rx}^{2}+V_{ry}^{2}}\right) \end{array}$$

According to the principle of interference, the condition for equal intensity of interference fringes at any two points *P*_1_ (*x*_1_, *y*_1_) and *P*_2_ (*x*_2_, *y*_2_) on the detector surface is that the difference in the optical path between the measurement light and the reference light at these two points is an integer multiple of the wavelength. Therefore, it can be concluded that
(16)$$\begin{array}{cc} \pm \frac{N\lambda }{n} & =\Delta L_{1}-\Delta L_{2}\\ & =(x_{1}-x_{2})\left(\frac{2V_{mx}}{1+V_{mx}^{2}+V_{my}^{2}}-\frac{2V_{rx}}{1+V_{rx}^{2}+V_{ry}^{2}}\right)\\ & +(y_{1}-y_{2})\left(\frac{2V_{my}}{1+V_{mx}^{2}+V_{my}^{2}}-\frac{2V_{ry}}{1+V_{rx}^{2}+V_{ry}^{2}}\right) \end{array}$$

In the equation, *N* is an integer, and Δ*L*_1_ and Δ*L*_2_ represent the path difference between the measurement light and the reference light at *P*_1_ and *P*_2_, respectively. Based on Equation (16), we can precisely determine the interference intensity at any point on the detector and analyze the interference fringes. Through the aforementioned modeling, it can be observed that variation in the interference signal’s pattern depends on the relative angle between the reference mirror and the measurement mirror. Although Equation (16) looks different from the expression of optical path difference in the level previously proposed by our team [23], it is essentially the same. According to Equation (16), when N = 1 and *y*_1_ = *y*_2_, the frequency in the x direction can be determined as follows:(17)$$f_{x}=\frac{1}{\Delta x}=\frac{n}{\lambda }\left(\frac{2V_{mx}}{1+V_{mx}^{2}+V_{my}^{2}}-\frac{2V_{rx}}{1+V_{rx}^{2}+V_{ry}^{2}}\right)$$
where *f_x_* is the spatial frequency of the interference fringes in the vertical direction, and Δ*x* is the fringe spacing in the horizontal direction. Similarly, the frequency in the *y* direction can be obtained. In this case, the angles of the level around the *X*-axis and the *Y*-axis can be determined as follows:(18)$$\left\{\begin{array}{c} \theta_{x}=\frac{\lambda f_{x}}{2n}\\ \theta_{y}=\frac{\lambda f_{y}}{2n} \end{array}\right.$$

### 2.2. High-Resolution-Angle-Decoupling Algorithm Based on Zoom FFT

From the results derived above, it can be seen that the frequency resolution obtained after solving the interference fringes in the proposed level directly determines the angle resolution. Therefore, the angle resolution of the level can be expressed as follows:(19)$$\Delta \theta =\frac{\lambda \cdot \Delta f}{2n}$$

In the frequency-domain transformation, the frequency resolution Δ*f* can be expressed as follows:(20)$$\Delta f=\frac{f_{s}}{N}$$
where *f_s_* is the sampling frequency, and *N* is the number of samples. According to Equations (19) and (20), we can derive the relationship between the angle resolution of the level, the size of camera pixels, and the number of pixels as follows:(21)$$\Delta \theta =\frac{\lambda \cdot f_{s}}{2n\cdot N}=\frac{\lambda }{2\cdot n\cdot N_{c}\cdot d_{pix}}$$
where Δ*θ* is the angle resolution, *d_pix_* is the size of camera pixels, and *N_c_* is the number of pixels. The CMOS camera used in the proposed level has a pixel size of 4.5 μm and a pixel count of 1600. The laser wavelength used is 632.8 nm. According to Equation (21), the calculated angle resolution is only 9 arcsec.

Based on Equations (20) and (21), it is known that improving the angle resolution can be achieved by reducing the sampling frequency and increasing the number of samples. However, in the proposed level, the sampling frequency and sampling number directly correspond to the number and size of the pixels of the CMOS camera, and these two parameters are fixed and cannot be changed when the CMOS camera is selected. Therefore, it is impossible to improve the angle resolution by directly reducing the sampling frequency and increasing the sampling number.

In the level previously proposed by our team [18], a zero-padding and curve-fitting algorithm was adopted. In this algorithm, the signal was first zero padded in the time domain, and then the padded signal was processed by means of FFT and fitting. Zero padding is equivalent to interpolating the signal in the frequency domain; thus, the frequency resolution was first improved, and then the angle resolution of the level was improved. However, increasing the number of interpolation points not only improves the resolution but also significantly increases the computational requirement of FFT and fitting, which occupies a large amount of computing resources. To achieve a more ideal angle resolution, it is necessary to increase the original data volume by hundreds of times and use a computer to solve the measured angle value offline. On the other hand, due to the limitation of circuit noise and the CMOS camera’s dark-field noise, the method becomes meaningless when the number of interpolation points is higher than a threshold. Therefore, the proposed level employs a decoupling algorithm based on zoom FFT in the signal processing unit to address this issue.

The signal processing module of the proposed level is shown in Figure 3. After detecting the interference fringes using the CMOS camera, the frequencies of the fringes in the x and y directions can be obtained. The process from obtaining the interference fringe to obtaining the frequency signal is shown in Figure 4.

The interference signal detected using the CMOS camera undergoes frequency shifting at first, which shifts the zero point of the signal spectrum to a frequency *f_e_*, thereby creating a new signal. *f_e_* is the frequency center of the frequency band of interest during the measurement process. For this level, the specific value of *f_e_* is often related to the initial deflection angle of the reference reflector. Before starting the measurement, the angle of the reference reflector needs to be adjusted to align the beams to generate the initial measurement interference fringes. Once the reference reflector is fixed, the value of *f_e_* is determined accordingly. The frequency band of interest during the measurement process manifests as the measurement range of the level. According to Equation (18), if the measurement range of the level is ±100 arcsec, the corresponding bandwidth of the interference signal is about 3.06 kHz.

After frequency shifting, the signal is low-pass filtered, and the cutoff frequency is denoted as *f_c_* and is half the bandwidth of the interference signal. After low-pass filtering, the interference signal contains only the frequency information within the measurement range of the level, while high-frequency information is removed. The filtered signal is then resampled. The processed signal has its frequency centered around zero, and the frequency resolution can be improved by reducing the sampling frequency. The resampling frequency is determined as follows:(22)$$f_{s}{}^{\prime }=\frac{f_{s}}{D}$$
where *D* is the multiple of refinement. According to the Nyquist sampling theorem, the sampling frequency should be at least twice the maximum frequency of the signal. Therefore, the relationship between the cutoff frequency of the low-pass filter and the sampling frequency is estimated using the following equation:(23)$$f_{c}\leq \frac{f_{s}}{2\cdot D}$$

After resampling, zero padding is performed to maintain the total number of samples consistent with before. Then, an N-point FFT transformation is applied to the resampled data. After combining with Equation (20), we can determine the frequency resolution at this time as follows:(24)$$\Delta f^{\prime }=\frac{f_{s}{}^{\prime }}{N}=\frac{f_{s}}{D\cdot N}$$

Next, FFT and frequency shifting are applied to the signal, and the frequency shift returns the signal to its original frequency. At this point, frequency refinement around the frequency *f_e_* of the signal is realized. Finally, a peak-search algorithm is used to identify the peak points in the signal, and the corresponding angles can be determined based on the frequency of each peak point. According to the above derivation process, it can be concluded that the angle resolution at this time can be expressed as
(25)$$\Delta \theta^{\prime }=\frac{\lambda \cdot \Delta f^{\prime }}{2n}=\frac{\lambda }{2n\cdot N_{c}\cdot d_{pix}\cdot D}$$

Figure 5 shows a comparison of the angle resolution obtained using the proposed algorithm and the zero-padding and curve-fitting algorithm, where *d_pix_* is 4.5 μm, *N_c_* is 1600, and *λ* is 632.8 nm. In the zero-padding and curve-fitting algorithm, the zero-padding operation on the signal is equivalent to increasing the number of sampling points. However, it can be seen from Equation (25) that the proposed decoupling algorithm can improve the angle resolution more effectively by changing the value of *D*, while keeping the number of sampling points unchanged.

In addition, according to the above derivation, it can be concluded that the measurement range of the level decreases with an increase in the refinement multiple (D). Figure 6 shows the relationship between angle resolution, measurement range, and the refinement multiple, where *d_pix_* is 4.5 μm, *N_c_* is 1600, and *λ* is 632.8 nm. When the value of D is set to 100 in the proposed level, the theoretical value of angle resolution is about 0.09 arcsec, and the measurement range of the level is only about ±72.5 arcsec. However, in an actual measurement scenario, the value of D is not set in stone and can be adjusted according to the measurement requirements.

## 3. Experimental Results

### 3.1. Functional Verification of Signal Processing Board

In order to show the improvement of the proposed algorithm, we tested the signal processing board and compared the results to those of previous zero-padding and curve-fitting algorithms. Simulated images with noise were given to the signal processing board. The variations of the interference fringes in the image were generated by a square-wave modulated angular motion simulation, with a period of 10 s and an amplitude of 0.1 arcsec. Random noise with a peak–peak value of 0.02 arcsec was added to it [3], and the signal duration was 60 s. The output signals of the signal processing board using different algorithms were stored separately, and the results are shown in the Figure 7. The experimental results show that when the proposed decoupling algorithm was adopted, the input signal could be clearly detected at a rate of 200 Hz, while the decoupling speed of the previous algorithm was below 0.1 Hz.

### 3.2. Stability Test of the Proposed Level

To verify the proposed level and its solution method, the resolution, repeatability, and accuracy were tested by using the level prototype. The experimental setup for the stability, repeatability, and resolution tests is shown in Figure 8. The experimental setup consisted of a 632.8 nm He–Ne frequency-stabilized laser, a single-mode polarization-maintaining fiber, a signal processing board, a high-precision nanopositioning stage with six degrees of freedom (6-DOF), and the level prototype. The He–Ne frequency-stabilized laser served as the light source and was connected to the level prototype via the single-mode polarization-maintaining fiber. The signal processing board was used to solve the interference signals received by the CMOS camera in the level. The 6-DOF high-precision nanopositioning stage was a P-562.6CD system manufactured by the German company PI, which was employed as the motion-generation device to drive the angle resolution down to 0.1 μrad.

The angle resolution of the level was first tested. Considering the influence of environmental noise and circuit noise during the test process, and in order to test the limit resolution of the proposed level, we set the refinement multiple D to 900. The nanopositioning stage was controlled to generate equally spaced round-trip steps of 0.1 arcsec, while recording the measurement results of the level and smoothing the obtained results by 10 points. The results are shown in Figure 9. It can be observed that the angle resolution of the proposed level was less than 0.1 arcsec.

### 3.3. Repeatability Test of the Proposed Level

Next, the repeatability of the proposed level was tested. The level was placed on the nanopositioning stage, and the drift of the system in one hour was less than 0.1 arcsec. On this basis, the repeatability test was carried out. The stage was controlled to generate steps of 20 arcsec along the *X*-axis with equal intervals. After recording the measurement results, the experimental setup was then adjusted, and the same procedure was repeated along the *Y*-axis. The final repeatability results of the level are shown in Figure 10. The test results indicate that the maximum repeatability accuracy error of the level was approximately ±0.2 arcsec, with a standard deviation of around 0.3 arcsec (*p* = 99.75%).

### 3.4. Angle Measurement Accuracy Test of the Proposed Level

The angle measurement accuracy of the proposed level was tested. In this study, a comparison test was performed to compare the level with a commercial autocollimator. The experimental setup is shown in Figure 11. The commercial autocollimator used in the experiment was an ELCOMAT^®^ 3000 autocollimator manufactured by Möller-Wedel, Wedel, Germany. It served as the angle reference device with a measurement resolution of 0.05 μrad, a measurement range of ±5 mrad, and a high-precision measurement accuracy of 0.5 μrad in the small central range. Before the experiment, the target reflectors of the autocollimator and the level were placed together on the nanopositioning stage. When the stage was deflected, both the level and the target reflector deflected simultaneously, and their deflection angles were the same.

Considering that in the precision comparison test, the proposed level needed to have a sufficient measuring range, the refinement multiple D was set to 100, and the corresponding measurement range was set to be about ±72.5 arcsec. The nanopositioning stage was controlled to generate steps of 6 arcsec with the same intervals, and a total of 10 steps were generated. The measurements of both the level and the autocollimator were recorded, as shown in Figure 12. In Figure 12, the two continuous curves represent the measurements of the level and the autocollimator, while the point plot represents the measurement error of the level relative to the autocollimator. To obtain this error, the average of the step portions in the measurement data of both devices was calculated, and the difference was taken. From the measurement results, it can be observed that the angle measurement accuracy of the level was approximately ±0.6 arcsec.

## 4. Conclusions

We proposed a two-dimensional precision level for real-time measurement, and a zoom-FFT-based decoupling algorithm was developed and integrated in an FPGA. Firstly, the principle of generating interference fringes is described in this paper, along with the process of detecting changes in the interference fringe frequency to obtain variations in the level’s angles. Secondly, the zoom-FFT-based decoupling algorithm is introduced. This algorithm surpasses hardware limitations by significantly enhancing the system’s frequency resolution through frequency-band selection, thereby improving the level’s angle resolution. Moreover, compared to the zero-padding and curve-fitting algorithms, this method reduces the number of calculation points, simplifies the calculation process, and makes real-time measurement possible. Finally, a series of experiments were conducted using a prototype to validate the performance of the proposed level. The principle analysis and test results showed that the angle resolution of the prototype could be improved from 9 arcsec to about 0.1 arcsec. The measurement repeatability of the proposed level was about ±0.2 arcsec and the angle measurement accuracy was about ±0.6 arcsec when compared to a commercial autocollimator.

In future research, the refinement multiple can be adjusted in real time by monitoring the measurement results of the level so that the measurement resolution and measurement range of the level can achieve a dynamic balance. Furthermore, the most critical factor affecting the measuring bandwidth of the level is the liquid-surface reflector. The material viscosity, depth, and liquid-surface size all affect the response speed as well as the bandwidth of the level. Smaller viscosity will result in higher bandwidth but will reduce the measurement resolution and stability. How to optimize measurement resolution and measurement range and achieve a compromise between the parameters of the liquid-surface reflector is a topic for follow-on research, and needs much more effort.

## Figures and Tables

**Figure 1 micromachines-14-02028-f001:**
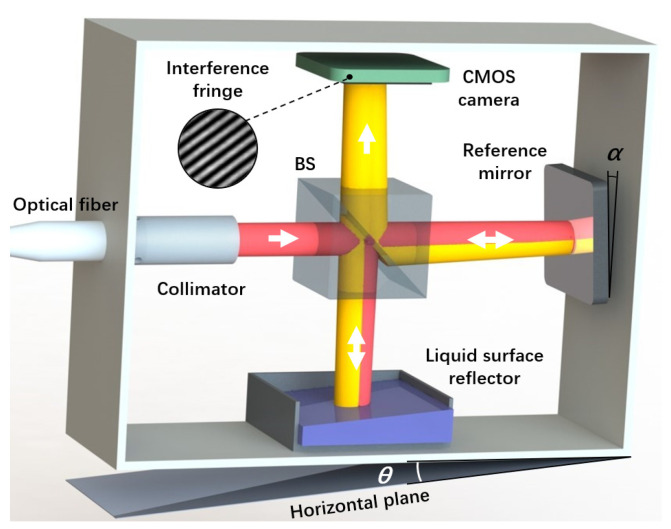
The structure of the two-dimensional precision level. BS, beam splitter; *α*, the initial deflection angle of the reference mirror; *θ*, the angle being measured.

**Figure 2 micromachines-14-02028-f002:**
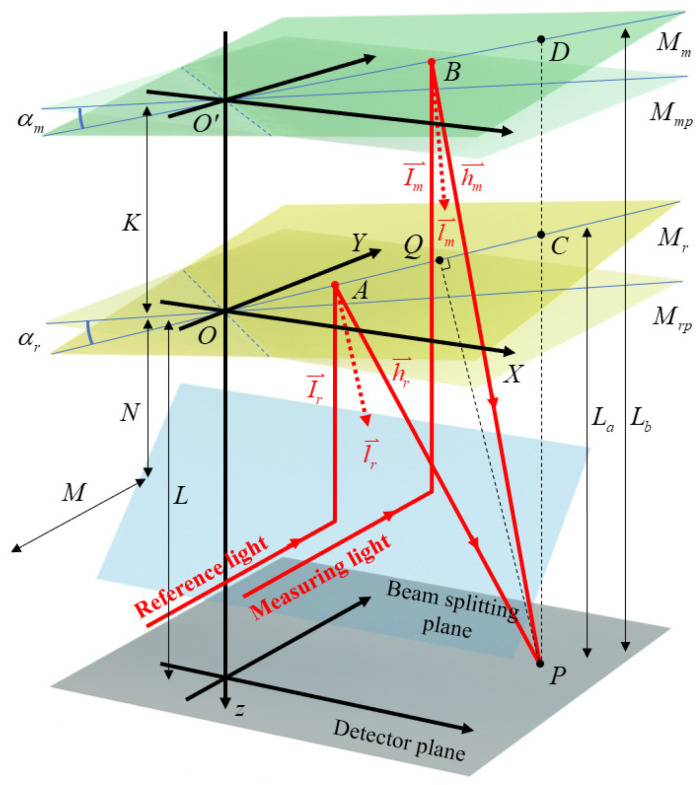
Illustration of the beam-tracing model. *α_m_*, the angle being measured; *α_r_*, the initial deflection angle of the reference mirror; *M_m_*, measurement mirror; *M_r_*, reference mirror; *M_mp_*, the ideal position of the measurement mirror; *M_rp_*, the ideal position of the reference mirror.

**Figure 3 micromachines-14-02028-f003:**
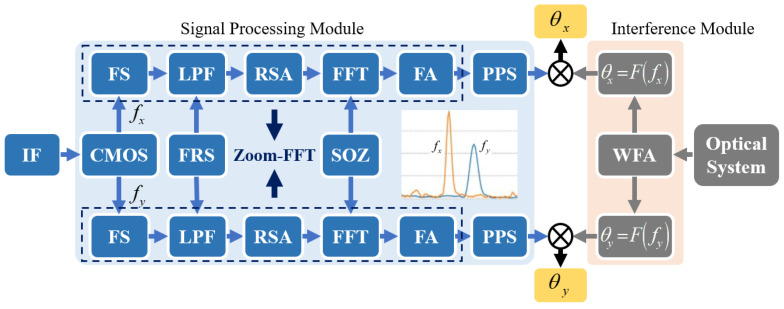
Signal flowchart of the proposed level. IF, interference fringe; FS, frequency shift; LPF, low-pass filtering; FRS, frequency range selection; RSA, resample; SOZ, sequence of zeros; FA, frequency adjustment; PPS, peak-point search.

**Figure 4 micromachines-14-02028-f004:**
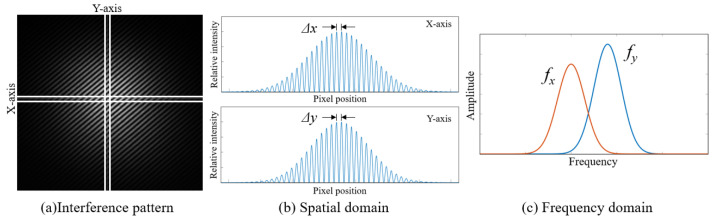
Characteristics of an interference pattern. (**a**) The interference fringe received by the CMOS camera, (**b**) the light-intensity curve of the interference fringe on the *X*- and *Y*-axes, and (**c**) the frequency-domain diagram of the interference fringes on the *X*- and *Y*-axes.

**Figure 5 micromachines-14-02028-f005:**
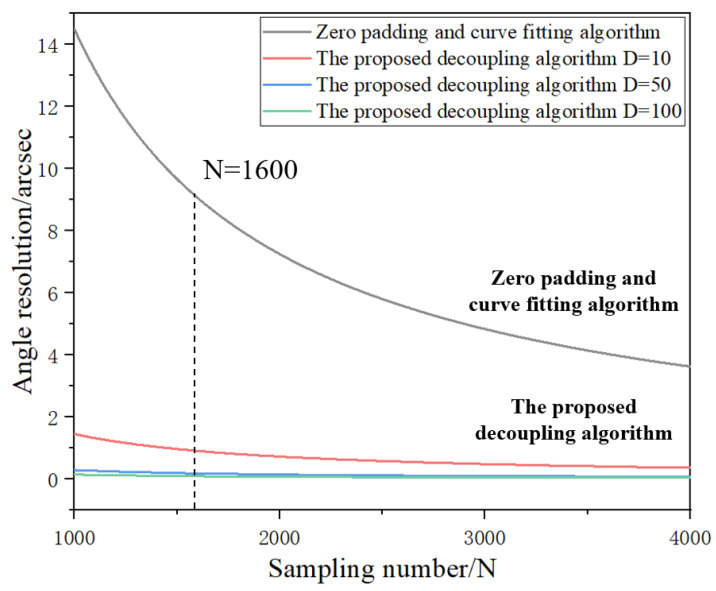
Comparison of angle resolution between the proposed decoupling algorithm and the zero-padding and curve-fitting algorithm. The *d_pix_* is 4.5 μm, *N_c_* is 1600, and *λ* is 632.8 nm.

**Figure 6 micromachines-14-02028-f006:**
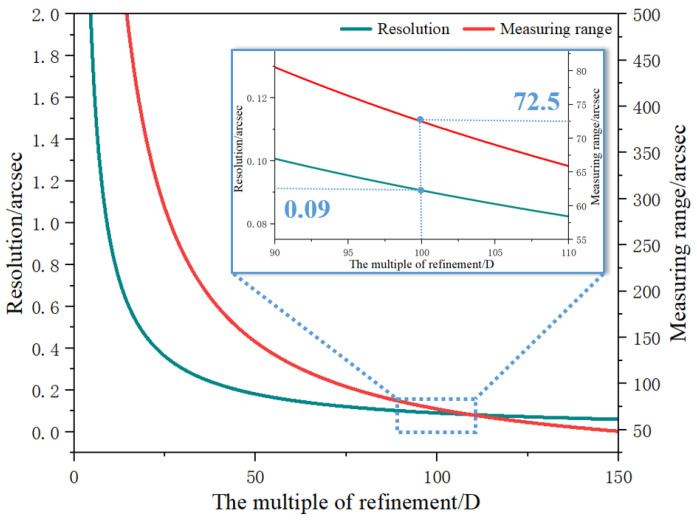
The relationship between angle resolution, measurement range, and the multiple of refinement (D). The *d_pix_* is 4.5 μm, *N_c_* is 1600, and *λ* is 632.8 nm.

**Figure 7 micromachines-14-02028-f007:**
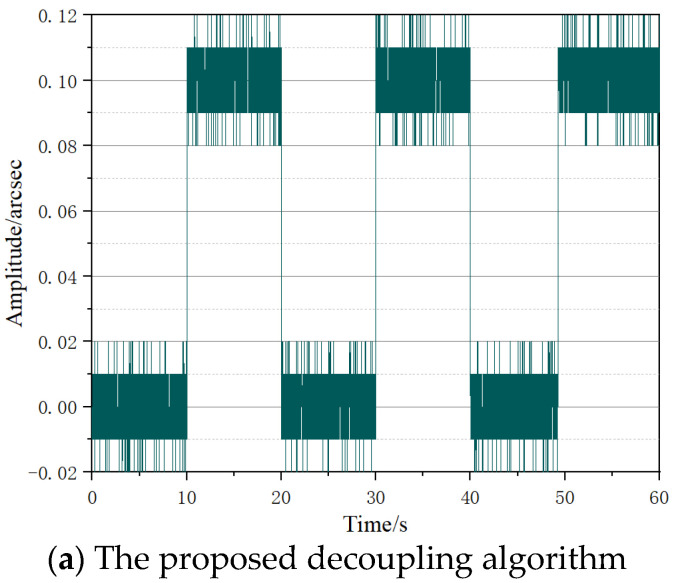
Simulation results of the signal processing board.

**Figure 8 micromachines-14-02028-f008:**
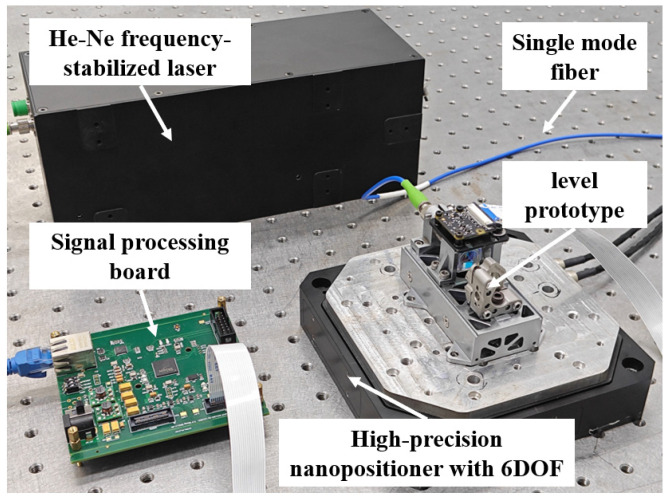
The experimental setup of the level.

**Figure 9 micromachines-14-02028-f009:**
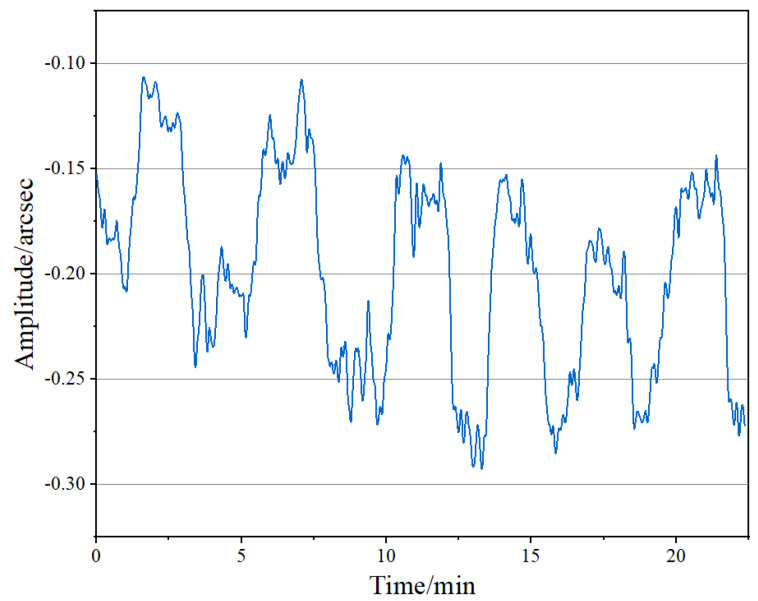
Angle resolution test results.

**Figure 10 micromachines-14-02028-f010:**
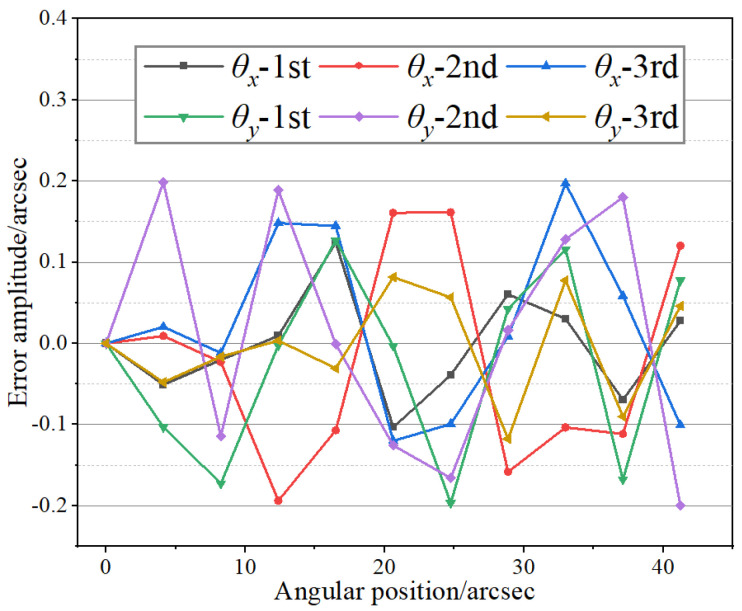
Repeatability test results.

**Figure 11 micromachines-14-02028-f011:**
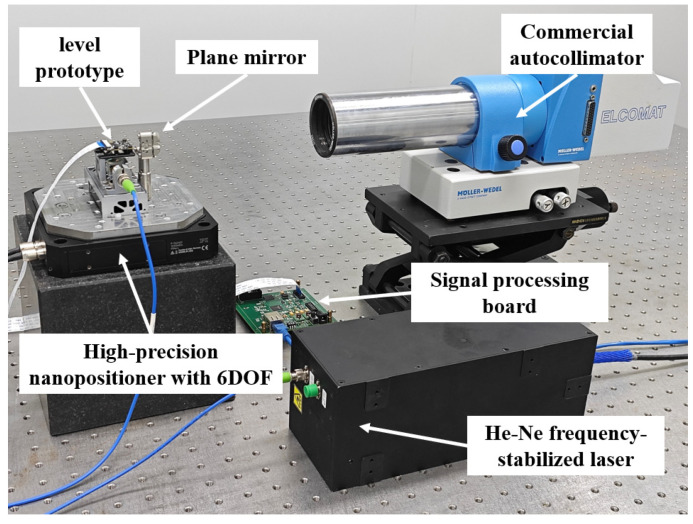
Experimental setup for angle measurement accuracy testing.

**Figure 12 micromachines-14-02028-f012:**
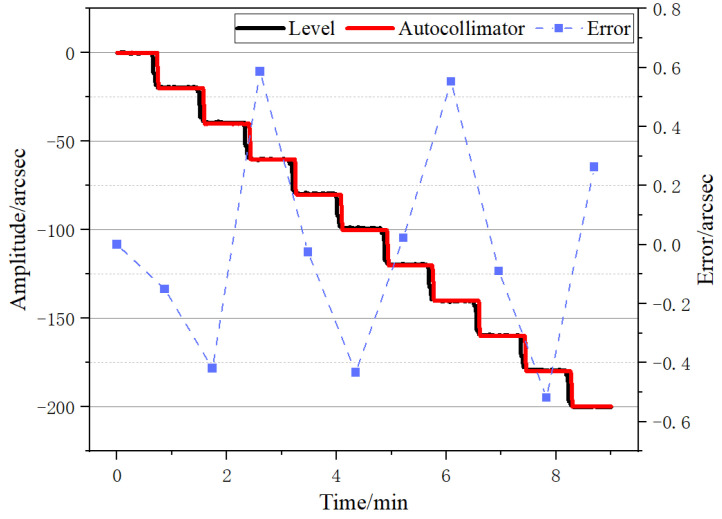
Angle accuracy test results.

## Data Availability

The data that support the findings of this study are available from the authors upon reasonable request.

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
