# Peer review of "A Two-Dimensional Precision Level for Real-Time Measurement Based on Zoom Fast Fourier Transform"

_micromachines, 2023, doi:10.3390/mi14112028_

Round 1

Reviewer 1 Report

Comments and Suggestions for Authors

It is a good paper and I want to see more on the topic and later research stages.

Author Response

Thanks for your comment and encouragement, we will make more efforts to carry out follow-up research!

Reviewer 2 Report

Comments and Suggestions for Authors

The authors of this paper propose a two-dimensional precision level using a Zoom-FFT based algorithm. Theory results of the proposed system are presented. Also, the experiments are conducted using a prototype to validate the performance of the proposed level. By implementing this strategy, the experiments have shown that the measured accuracy of angle is about ±0.6 arcsec compared with a commercial autocollimator.

After reading through this paper in its current form I believe that this manuscript requires improvements and revisions before it can be accepted. Interesting results were presented. However, it needs to be further developed to qualify for publication. Please see my comments below:

1.         The heart of the paper is focused on real-time measurement of the level, but the experimental results is not convincing. Although in Figure 12 the amplitude changes over time, the signal actually changes in steps. In addition, you should give the measured bandwidth of the level.

2.         The abstract should be refined.

3.         The figure captions such as in fig. 1 and Figure 2 are unclear. You need to describe in detail.

4.         Please explain the “Frequency Range Selecshuotion” in Figure 3.

5.         Please give the parameters of the theoretical results in Figure 5 and 6 both in the text and the figure captions. For example, the pixel size and wavelength used in the fitting.

6.         How do you define the accuracy error of the level in Figure 10?

7.         The english – structure and grammar –needs to be further refined/improved throughout the paper.

8.         The quality of the figures should be improved.

Comments on the Quality of English Language

 The english needs to be further refined/improved throughout the paper.

Reviewer 3 Report

Comments and Suggestions for Authors

This manuscript presents a two-axis level which can perform real-time measurement. A Zoom-FFT based algorithm is developed to eliminate the contradiction between the measurement speed and resolution of the level. This is meaningful work, which efficiently improves the practicality of a novel optical level. The structure of the manuscript is reasonable, the content is substantial and the experiment is detailed. Therefore, I suggest to accept the manuscript after the following issues are addressed:

1)In the introduction of the manuscript, it is pointed out that other angle measurement methods based on interference fringes also have contradictions between the measurement resolution and the measurement speed. The question is, does the proposed algorithm also fits these methods? Is it generally applicable or only useful for the proposed level?

2)Most of the existing methods of angle measurement based on interference fringes use a plane mirror as a reflection device. However, in the proposed level, part of the light is reflected by liquid surface (with only ~3% reflection rate), the intensity of the interference light will be much weaker than usual interferometers, how does it influence the proposed level, e.g. power consumption and ghost reflection? Is it possible to use mirror and beam splitter with different reflection ratio to save some light power? Moreover, it can be seen from the photos that the prototype is small in size. In this case the liquid surface may not be used as an ideal plane, but a curve. How does this affect the level?

3)The curve of the relationship between the measuring range of the level and the refinement multiple D is pointed out in page 9 of the manuscript, but the authors did not explain why the refinement multiple D is limiting the measuring range. How does it work?

Comments on the Quality of English Language

None

Round 2

Reviewer 2 Report

Comments and Suggestions for Authors

The Authors responded to my comments and suggestions. Therefore, I suggest accepting the paper for publication.